# The Roles of CDPKs as a Convergence Point of Different Signaling Pathways in Maize Adaptation to Abiotic Stress

**DOI:** 10.3390/ijms24032325

**Published:** 2023-01-24

**Authors:** Hanwei Du, Jiajia Chen, Haiying Zhan, Shen Li, Yusheng Wang, Wei Wang, Xiuli Hu

**Affiliations:** State Key Laboratory of Wheat & Maize Crop Science, College of Life Sciences, Henan Agricultural University, Zhengzhou 450002, China

**Keywords:** maize, CDPKs, signal transduction, abiotic stress

## Abstract

The calcium ion (Ca^2+^), as a well-known second messenger, plays an important role in multiple processes of growth, development, and stress adaptation in plants. As central Ca^2+^ sensor proteins and a multifunctional kinase family, calcium-dependent protein kinases (CDPKs) are widely present in plants. In maize, the signal transduction processes involved in ZmCDPKs’ responses to abiotic stresses have also been well elucidated. In addition to Ca^2+^ signaling, maize ZmCDPKs are also regulated by a variety of abiotic stresses, and they transmit signals to downstream target molecules, such as transport proteins, transcription factors, molecular chaperones, and other protein kinases, through protein interaction or phosphorylation, etc., thus changing their activity, triggering a series of cascade reactions, and being involved in hormone and reactive oxygen signaling regulation. As such, ZmCDPKs play an indispensable role in regulating maize growth, development, and stress responses. In this review, we summarize the roles of ZmCDPKs as a convergence point of different signaling pathways in regulating maize response to abiotic stress, which will promote an understanding of the molecular mechanisms of ZmCDPKs in maize tolerance to abiotic stress and open new opportunities for agricultural applications.

## 1. Introduction

Abiotic stresses in nature mainly include temperature, salt, drought, and water stress, etc., which have adverse effects on plant growth [1]. Plants have formed complex regulatory mechanisms to respond to these environmental changes, particularly through transmitting signals. The second messenger, Ca^2+^, is an important element in signal transduction. The Ca^2+^ signal system is one of the common intracellular signal transduction pathways in plants. Ca^2+^ can sense almost all external stress signals and is responsible for subsequent signal transduction [2]. When plants are stimulated by external stimuli, Ca^2+^ concentration will change transiently in cells, cascade amplifying the signal and transmitting it to the intracellular level, and then these changes can be recognized and perceived by specific calcium sensors/receptors, thereby inducing further response processes [3].

A variety of protein kinases in cells are the targets of the above intracellular messengers, which further transmit information by phosphorylating intracellular proteins and regulating their activities. Calcium-dependent protein kinases (CDPKs) are a kind of calcium sensor with a calcium binding domain and a kinase domain [4], which can directly convert calcium signals into phosphorylation events [5]. CDPKs are Ca^2+^-regulated serine/threonine protein kinases [6] that are present in plants, protozoa, oomycetes, and green algae, rather than in animals and fungi [7], and they are composed a large family of protein kinases [8], which are involved in the response to various abiotic stresses in plants [9]. CDPKs have been identified in many plants, such as Arabidopsis, maize, rice, tobacco, potato, and so on. CDPKs are distributed in most tissues, such as leaves, roots, flowers, fruits, etc., and some CDPKs are specifically expressed [10,11,12]. Multiple CDPKs are subcellularly localized on the plasma membrane, which is determined by the myristoylation site of their related N-terminal variable domain [13]. CDPKs also exist in other subcellular locations, such as cytosol, nucleus, endoplasmic reticulum, peroxisome, mitochondria, chloroplasts, etc., and some will change their localization with the occurrence of stress [14,15,16,17,18], suggesting that CDPKs play roles in a variety of physiological processes [6]. CDPKs mainly interact with other proteins in a calcium-dependent way, exert their kinase activity, and phosphorylate the Ser/Thr sites of target proteins to regulate their activity, thereby regulating the process of plant signal transduction. CDPKs, as central regulators of Ca^2+^-mediated stress response or as hubs, affect the survival status of plants and have important biological functions in plant response to abiotic stress.

In maize, 40 CDPKs have been identified [19]. Most of these CDPKs are regulated by Ca^2+^ and can respond to a variety of stimuli, including salt, drought, cold, heat, abscisic acid (ABA), and hydrogen peroxide (H_2_O_2_) [20,21]. More and more studies have found that CDPKs play the role of signal transmission in the process of maize adaptation to abiotic stress, can sense the changes of molecular signals caused by environmental stress, and transmit these changes to different types and functions of downstream proteins, thus participating in multiple signal transduction pathways. This review summarizes CDPKs as the convergence point of different signaling pathways in maize, which can respond to abiotic stress by activating downstream target proteins.

## 2. Activation and Expression of ZmCDPKs under Abiotic Stress

Up to the present, 40 CDPKs have been found in maize [19] (Table 1), which are widely distributed in different tissues and organs and have various functions. Among them, multiple maize ZmCDPKs play a key role in abiotic stress [22]. Most ZmCDPKs function in a calcium-dependent manner. Many studies have shown that the regulation of ZmCDPKs by Ca^2+^ is mainly determined by the protein structure of ZmCDPKs. The maize ZmCDPKs contain the following four domains: the amino terminal variable region containing myristoylation and/or palmitoylation sites; the Ser/Thr kinase functional region; the junction region; and the calmodulin-like domain [23]. In the absence of Ca^2+^ stimulation, the junction region bound to the kinase functional domain acts as a pseudo-substrate to inhibit the kinase activity; in the presence of Ca^2+^, Ca^2+^ can bind to the calmodulin-like domain, which changes the protein structure, and it then drives the junction region from the protein kinase domain; thus, ZmCDPKs are activated, and the protein kinase domain is released, which then binds to its substrate and performs phosphorylation function to mediate Ca^2+^ signal transduction [24,25,26]. However, in the current study, it was found that not all ZmCDPKs were induced by Ca^2+^, and the activation and function of some ZmCDPKs may not depend on Ca^2+^. For example, the expression of *ZmCPK26* and *ZmCPK28* did not change significantly under Ca^2+^ treatment [27], indicating that they were not sensitive to Ca^2+^ treatment. 

In addition to Ca^2+^, the expression of some *ZmCDPKs* in maize was enhanced by multiple stresses. *ZmCPK35/37* was significantly induced under drought stress; *ZmCPK11* could be induced not only by drought and salt, but also by ABA and H_2_O_2_; but *ZmCDPK12* could only be induced by drought and salt. The expression of *ZmCPK6/7* was significantly upregulated by heat stress [23,30]; and the expression of *ZmCDPK4* was upregulated by cold and drought stress. In addition, salt stress upregulated the transcription levels of *ZmCPK3*, *ZmCPK14*, *ZmCPK28,* and *ZmCPK37* [19]; drought stress upregulated *ZmCPK1*, *ZmCPK11*, *ZmCPK22*, *ZmCPK28,* and *ZmCPK39*; and cold stress upregulated the expression of *ZmCPK14*, *ZmCPK17,* and *ZmCPK31* [19]. Interestingly, the expression of *ZmCDPK1* and *ZmCDPK25* was significantly induced under cold stress conditions, but these proteins negatively regulated cold stress tolerance [28]. By contrast, the expression of some *ZmCDPKs* was downregulated under abiotic stress; for example, the expression of *ZmCPK5/29* and *ZmCPK33* was decreased by salt stress, which may have negative regulation on salt stress. This also partly showed that not all ZmCDPKs are positively regulated in response to stress. In addition, the expression of multiple *ZmCDPK* genes in maize was also induced by ABA and hydrogen peroxide, such as *ZmCDPK7*, *ZmCDPK11*, *ZmCDPK17,* and *ZmCPK39* [19,27]. At the same time, the expression of some *ZmCDPKs* was decreased after ABA treatment, such as *ZmCPK5*, *ZmCPK11,* and *ZmCPK33* [19]; hydrogen peroxide treatment inhibits *ZmCPK4* expression [34]. The expression of some *ZmCDPKs* was affected by more than one stress, but their functions have not been elucidated [19].

## 3. ZmCDPKs Activate Transporters to Start Ion transport

Transport proteins are responsible for mediating the exchange of signal and chemical substances inside or outside the plasma membrane, and they are important to signal transduction [35]. Ion channel protein is also a transport protein. When plants are subjected to drought stress, high temperature, high CO_2_ concentration, pathogen invasion, and other stresses, stomatal aperture will decrease to regulate the dynamic balance of leaf transpiration and photosynthesis to improve the survival ability of plants [36,37,38]. In plants, the ion channel located in guard cells is a key component of ion transport, which regulates the stomatal opening state by regulating the turgor pressure changing of guard cells via ion transport [39]. Studies have shown that CDPKs can regulate stomatal opening by regulating these ion channel proteins and thus play a capital role in plant response to stress. For example, an ABA-regulated S-type anion channel protein, SLAC1, could regulate stomatal closure by mediating anion (mainly, NO_3_ and Cl^-^) efflux to further respond to stress [40,41]. SLAC1 was also activated by CDPK-mediated phosphorylation events [42]. AtCPK6 could interact with SLAC1 in Xenopus oocytes and regulate the anion transport activity through phosphorylation, which further mediated stomatal closure under drought conditions [43]. In addition to AtCPK6, CPK3/21/23 can also activate this ion channel [44,45,46], thus enhancing plants’ stress adaptation. In maize, a similar anion channel was activated in this way. ZmSLAC1, located in guard cells, could be preferentially expressed in maize under drought stress. ZmSLAC1 is also an S-type anion channel, similar to Arabidopsis AtSLAC1 [47], which can protect anion efflux under stress conditions and mediate stomatal closure. Two maize calcium-dependent protein kinases, ZmCPK35 and ZmCPK37, which are located in guard cells, could interact with ZmSLAC1 and activate ZmSLAC1 to protect anion efflux in plant cells and thus promote maize stomatal closure under drought conditions (Figure 1). *ZmCPK37* overexpression/mutants in maize respectively increased/decreased plant tolerance to drought stress, indicating that ZmCPK37 transmits stress signals to the anion channel ZmSLAC1 by phosphorylation, which further regulates osmotic balance to improve drought tolerance in maize [29]. Moreover, ZmCDPK4 was also involved in ABA-mediated stomatal closure under drought stress. Under ABA treatment conditions, compared with a wild type, overexpression of *ZmCDPK4* in *Arabidopsis thaliana* significantly reduced the dehydration rate of leaves under drought conditions and significantly enhanced the drought tolerance of *Arabidopsis thaliana*, indicating that ZmCDPK4 may respond to drought stress by mediating stomatal closure [34]. Under drought conditions, the process of CDPKs mediating ABA-regulated stomatal closure is likely to be achieved through the activation of ion channels, but the issue of whether ZmCDPK4 regulates ion channel proteins in mediating maize response to drought stress is not investigated in current research (Figure 1) [34].

When the Na^+^ ion content increases to a high concentration, it will be toxic to plants and induce senescence or even necrosis of plant tissues [48,49]. Na^+^ transport proteins can reduce the accumulation of toxic Na^+^ in plant cells, which is an important mechanism of salt tolerance in plants. K^+^ improves plants’ tolerance to various stresses [50]. Excessive Na^+^ inhibits the activity of many enzymes that require K^+^ to function. Therefore, the survival of plants under salt stress requires a high K^+^/Na^+^ ratio in the cytoplasm [51]. CDPKs, as calcium sensors, can also sense changes in K^+^/Na^+^ ratio and function in K^+^/Na^+^ steady-state regulation [50]. Studies have shown that Na^+^/H^+^ antiporters in plasma membrane, the SALT OVERLY SENSITIVE 1 (AtSOS1), high-affinity potassium transporters (AtHKT1), Na^+^, K^+^, H^+^ antiporters, and Na^+^/H^+^ exchanger 1 (AtNHX1) are important Na^+^/H^+^ antiporters in Arabidopsis, which play an important role in salt tolerance by regulating the content of Na^+^ and K^+^ [52]. The activity of these ion transport proteins was regulated by maize calcium-dependent protein kinases [53]. Overexpression of *ZmCDPK11* in Arabidopsis could increase the expression of *AtSOS1*, *AtHKT1,* and *AtNHX1* under salt stress conditions. Moreover, the K^+^/Na^+^ ratio was higher in *ZmCDPK11*-overexpressing Arabidopsis than in WT [31]. This suggests that ZmCDPK11 regulates the K^+^/Na^+^ ratio in plants by regulating the expression of *AtSOS1*, *AtHKT1,* and *AtNHX1* genes, thereby responding to salt stress (Figure 1). In addition, grapevine VaCDPK20/21 has also been shown to regulate AtNHX1 in response to cold and drought stress [54,55]. However, whether the related transporter proteins in maize are activated through phosphorylation by ZmCDPKs has not been studied.

## 4. ZmCDPKs Activate Heat-Shock Proteins to Stabilize Protein Function

Heat-shock proteins (HSPs) are important molecular chaperones [56], which can be rapidly synthesized in large quantities when plants encounter stress signals such as heat, drought, oxidation, cold, and ABA treatment, and they help to restore the activity of misfolded proteins after stress. HSPs are often involved in the protein nascent peptide chains’ transport, folding, assembly, localization, and other functions in cells [57,58]. Heat-shock factors, HSFs, are a transcription factor that regulates HSPs’ transcription [59]. HSPs can be activated and exert molecular chaperone activity via phosphorylation [58,60,61]. HSPs could be phosphorylated by CDPKs. CPK10(AtCPK1) could bind to HSP1 and regulate drought resistance of Arabidopsis [62]. AtCPK3 and AtCPK13 phosphorylated heat shock factor HSFB2a [63], which could be an indirect regulation of HSPs. In maize, ZmCDPKs can activate some HSPs, thereby regulating the stress tolerance of maize. For example, the plasma membrane-localized protein kinase, ZmCDPK7, was induced by heat stress, and its localization was transferred to the cytoplasm when stress occurs, and it then interacted with the small heat-shock protein sHSP17.4, which was located in the cytoplasm, and phosphorylated it, activating its molecular chaperones’ activity [30]. Moreover, ZmCDPK7 could also interact with HSP70. However, the specific regulatory mechanism remains to be further studied. In addition, the expression levels of *sHSP17.4* and *HSP70* could also be regulated by ZmCDPK7 (Figure 1). *ZmCDPK7* overexpression/knockout up- and downregulated the expression of *sHSP17.4* and *HSP70* in maize under heat stress conditions, respectively. Maize plants with respective *ZmCDPK7* overexpression/mutant showed more tolerance/sensitivity to heat stress, further illustrating that ZmCDPK7 regulated the heat tolerance of maize through phosphorylating the substrate HSPs [30].

## 5. ZmCDPKs Activate Transcription Factors to Mediate Gene Expression

Transcription factors (TFs) are essential for plant responses to abiotic and biotic stresses [64,65]. TFs are responsible for binding to the promoters of downstream genes, thereby activating their transcription to express corresponding proteins. When stress occurs, a large number of proteins in plants need to be expressed instantaneously to cope with stress, and at this time, TFs are extremely important. TFs also need to be activated by phosphorylation. Multiple TFs can respond to stress, and studies have found that some TFs are regulated by CDPKs. CDPKs in maize also seemed to activate some TFs, such as ZATs of the C_2_H_2_ zinc finger protein family TFs, CBFs of the AP2/EREBP family, and ABFs and ABI5 of the bZIP family, which are regulated by Arabidopsis AtCPK1 to cause signal transmission. These family TFs are related to plant responses to stress. ZmCPK11 might regulate ABA and salt-induced signaling pathways by directly binding to candidate TFs and affecting their activity [32]. In Arabidopsis with *ZmCPK11* overexpression, the expression of *AtZAT6*, *AtZAT10*, *AtCBF1*, *AtCBF2,* and *AtCBF3* genes was upregulated [31]. AtZAT6/10 belongs to the C_2_H_2_ zinc finger protein family. Studies have shown that AtZAT6 can bind to the promoter of transcription factor AtCBFs, activate their transcription, and participate in plants’ response to salt, drought, and cold stress [66]. AtZAT6 and AtCBFs were also regulated by ZmCDPK11, indicating that ZmCDPK11 was likely to transmit signals to AtCBFs by phosphorylating ZAT6, thereby regulating plants’ response to environmental stress [31]. 

Transcription factor ABFs (ABA-responsive element binding factors) and ABI5 (abscisic acid-insensitive protein) belong to the basic leucine zipper proteins family, or the bZIP transcription factor family. They can specifically recognize the ABA-responsive element (ABRE), priming the transcription of stress-related genes. As such, they are important factors in the ABA signal transduction pathway [67,68]. ABF1 and ABF4 were phosphorylated by AtCPK4/11 in Arabidopsis. Similar regulation was also found in maize. In *Arabidopsis thaliana* with overexpressing maize *ZmCDPK4*, the expression of *ABF3* and *ABI5* were significantly upregulated, indicating that ZmCDPK4 plays a positive role in the ABA signal transduction pathway under abiotic stress in plants. Interestingly, ZmCPK1 and ZmCPK25 in maize can negatively regulate cold stress by inhibiting transcription factor ZmERF3. The analysis of *ZmCPK1* and *ZmCPK25* promoters in maize showed that there was a cis-acting element LTRE, which is necessary for cold stress-induced gene expression [69]. This determined that the gene expression of both was sensitive to cold. ZmERF3 is identified as a cold stress response marker gene, which is activated by cold stress [70]. ZmCPK1 inhibited the expression of *ZmERF3*. Moreover, ZmCPK1 directly phosphorylated ZmERF3 to inactivate it. This indicated that ZmCPK1 has a negative regulatory effect on cold stress [28]. There are many classic TFs that respond to stress and are regulated by CDPKs. For example, WRKY family transcription factors AtWRKY8/28/48 are activated by AtCPK4/5/6/11 [71], NAC family transcription factor OsNAC45 can be induced by OsCPK9 in rice [72], and AP2/EREBP family transcription factors AtDREB1A/2A can be induced by grapevine VaCPK29 [73]. However, there are few studies on whether these family transcription factors are directly regulated by CDPKs in maize.

## 6. ZmCDPKs Involved in Plant Hormone-Signaling Response to Stress

Plants respond to stress by regulating hormone levels. Plant hormones play an indispensable part in plant adaptation to environmental changes [74]. There is a very complex relationship between CDPKs and plant hormone signals. Abscisic acid (ABA) is a vital hormone that is well known to be involved in a variety of abiotic stress responses and tolerance [75], is involved in multiple signaling pathways in plants’ response to abiotic stress, and has crosstalk with Ca^2+^ signaling [25]. A large number of studies have shown that ABA and CDPKs have synergistic effect in plant response to abiotic stress. For example, CPK4 and CPK11 may be two positive regulators in the calcium-mediated ABA signaling pathway [76]. AtCPK10 is involved in plant stomatal movement regulated by ABA and Ca^2+^ under drought stress [77]. Similar to ABA, the JA signaling pathway also regulates plant stress response [78] and is closely related to the Ca^2+^ signal transduction pathway and Ca^2+^ sensor CDPKs [21]. For example, under salt stress, the ABA and JA signaling pathways were activated, which was followed by the inhibition of root length [78,79]. The plant hormones’ response to stress can affect the growth and development of plants, reflecting the cross-integration of signals between plant abiotic stress response and plant growth and development, which is related to hormones [80]. Other hormones are also closely related to CDPKs’ signal, but deep research is needed.

ABA can upregulate the expression of several CDPKs in maize, among which ZmCDPK7 and ZmCDPK11 have been analyzed in detail. However, the expression of *ZmCDPK4* was downregulated under ABA treatment. Studies have shown that overexpression of *ZmCDPK4* in Arabidopsis enhanced the maize sensitivity to ABA. ABA treatment resulted in leaf yellowing of Arabidopsis with overexpressing maize *ZmCPK4* [34]. The promoter sequence of ZmCDPK7 response to heat stress in maize contains the ABA response element ABRE. Under heat stress, the expression of *ZmCDPK7* in the ABA deletion mutant *vp5* was significantly lower than that in its wild type *Vp5*, and the exogenous application of ABA and its inhibitor sodium tungstate enhanced and decreased the expression of *ZmCDPK7* under heat stress conditions, respectively. This indicated that the expression of *ZmCDPK7* is regulated by ABA [30].

In addition to *ZmCDPK7*, ABA significantly induced the expression and activity of *ZmCPK11* under water stress [32]. ZmCPK11 participated in the response of ABA and JA signals to salt stress and improved the salt tolerance of *Arabidopsis thaliana*. Under salt stress, the expression of *ZmCPK11* was upregulated, and ZmCPK11 had sharp kinase activity. It was found that overexpression of *ZmCPK11* in *Arabidopsis thaliana* was beneficial to root development under normal conditions, but inhibited root growth under salt stress, which was affected by ABA and JA levels. MeJA could also upregulate expression of *ZmCPK11* and the kinase activity of ZmCPK11 [81]. Under normal conditions, the JA level of *ZmCPK11* overexpression plants was significantly lower than that of the wild type, and under salt stress, the JA level was higher than that of the wild type. In overexpression lines, the ABA level under normal conditions and salt stress conditions was also consistent with the JA level. After exogenous application of ABA or MeJA, the root length of Arabidopsis with *ZmCPK11* overexpression was shorter than that of the wild type, indicating that ZmCPK11 inhibiting the root growth under salt stress may be due to changes in hormone signals [31]. In addition, the ABA response element ABRE was also found in the promoter of drought stress response gene *ZmCDPK12* [33], suggesting that the gene may also be involved in the ABA signaling pathway.

In addition to being directly involved in plant hormone response to stress signals, CDPKs are also involved in the regulation of hormone signal-related genes. For example, AtZAT6, a transcription factor regulated by ZmCDPK11, is a key regulator of ABA and JA signaling [31]. Other transcription factors regulated by ZmCDPK11, such as AtZAT10 and AtCBFs, have also been shown to be regulated by JA signaling [82,83]; the transporter NHX1 was regulated by ABA signaling [84]. Overexpression of *ZmCDPK4* in Arabidopsis significantly upregulated the ABA-induced drought stress response gene *KIN1/KIN2* in the presence of ABA. ZmCDPK11 could regulate the expression of *MAPK5* under the action of ABA signal under stress [32]. More importantly, ZmCDPK4 can respond to stress through the key proteins ABF3 and ABI5 of the ABA signaling pathway [34]. These two types of proteins are well-known ABA regulators, and their interactions with CDPKs have also been studied frequently. For example, AtCPK7 in Arabidopsis regulated ABF1/2 [4]; Grapevine VaCPK1/20/26 regulated ABF3 [54,85]; Arabidopsis AtCPK12 regulated ABF1/4; and ABI2 [86]; AtCPK21/23 regulated ABI1 [44]. Taken together, these results indicated that CDPKs are closely related to key proteins in the ABA signaling pathway, and the hormone-signaling pathway-related proteins of CDPKs’ downstream systems are worthy of attention.

## 7. ZmCDPKs Involved in ROS Signaling Response to Stress

The production of CDPK-mediated reactive oxygen species (ROS) is required for multiple signaling pathways in plants. When plants respond to stress, a large amount of ROS, such as O_2_^−^, H_2_O_2_, and ^1^O_2_, are accumulated. They have a variety of biological effects and are essential for plants’ stress response as signaling molecules. Among them, H_2_O_2_ treatment can improve the heat tolerance of plants [87]. An important source of ROS is produced by respiratory burst oxidase homologue RBOHs [88]. Existing research has shown that RBOHs are involved in heat stress, osmotic stress, and salt stress response [89,90,91]. There have been 13 RBOHs identified in maize, including ZmRBOHA-M [92].

ROS produced by RBOHs could affect the expression of *ZmCDPK4*, *ZmCDPK7,* and *ZmCDPK11* and in turn regulate them [30,32,34]. H_2_O_2_ treatment could significantly increase the expression of *ZmCDPK4*. The pretreatment of NADPH oxidase inhibitor DPI decreased the expression of *ZmCDPK7* in maize under heat stress [30]. The expression of *ZmCDPK11* was induced by H_2_O_2_. ABA could induce the production of H_2_O_2_ and the expression of *ZmCDPK11* by a H_2_O_2_-dependent way in maize. The pretreatment with DPI and catalase (CAT) significantly inhibited the expression of ABA-induced *ZmCDPK11* in maize [32]. These studies indicate that ROS can induce the expression of *ZmCDPK4*, *ZmCDPK7,* and *ZmCDPK11* in maize, and ZmCDPKs are widely engaged in the ROS signaling pathway.

On the other hand, ZmCDPKs could also regulate the production of ROS. ZmCDPK7 could increase the expression of *ZmRBOHs* under heat stress. RBOHs needs to be activated via phosphorylation, which seems to be regulated by CDPKs [93]. For example, Arabidopsis AtRBOHD could be rapidly phosphorylated by AtCPK5 in a short time, and the activated AtRBOHD could mediate ROS production and thus participate in stress responses [94]. In addition, constitutively activated AtCPK1/2/4/11 could phosphorylate the N-terminus of AtRBOHD [95]. Moreover, CDPKs in rapeseed, potato, tobacco, etc. are also related to the phosphorylate activation of RBOHs [20,96,97]. This shows that CDPKs play an important role in the regulation of ROS signal in plants. This regulation also exists in maize. Under heat stress conditions, the relative expression levels of *ZmRBOHs* were increased/decreased in *ZmCDPK7* overexpression/knockout maize plants, respectively. This effect by *ZmRBOHB* is most pronounced. Moreover, ZmCDPK7 could phosphorylate ZmRBOHB to mediate ROS production and regulate maize tolerance to heat stress (Figure 1) [30]. In plants, excessive ROS may cause membrane lipid peroxidation, protein denaturation, carbohydrate oxidation, pigment decomposition, and DNA damage [98]. Therefore, timely clearance of ROS is critical in plants for survival through environmental changes [99]. Antioxidative defense enzymes, such as superoxide dismutase (SOD), ascorbate peroxidase (APX) and catalase (CAT), can effectively remove excess ROS [100,101,102]. Under heat stress conditions, overexpression/knockout of *ZmCDPK7* in maize could significantly increase/decrease the expression of *ZmAPX1* and *ZmCAT1* and increase/decrease the enzyme activities of APX and CAT. Correspondingly, the content of heat-induced H_2_O_2_ was decreased/increased significantly [30]. In the study of maize ZmCDPK11, the expression of *ZmSOD4* and maize chloroplast *ZmcAPX* and the activities of SOD and APX increased significantly after transient overexpression of *ZmCDPK11*, whereas transient silencing of ZmCDPK11 showed the opposite result [32]. In addition, ZmCDPK11 could also indirectly regulate ROS production by upregulating the expression of *AtZAT6,* which can respond to stress by regulating ROS levels. Moreover, ZmCDPK11 could enhance SOD and APX activities by upregulating *ZmMAPK5* expression in maize [66]. These studies show that ZmCDPK7/11 is important in maintaining the homeostasis of ROS in maize response to stress.

## 8. ZmCDPKs Interact with Other Protein Kinases

The mitogen-activated protein kinase family (MAPK) is an important signal transduction centrum. When plants respond to abiotic stress, they transmit signals through phosphorylation, which triggers transduction cascades and regulates gene expression and other processes [103]. MAPKs are regulated by ABA [104]. The MAPK protein has a very classic signal transduction pathway, the MPKKK-MPKK-MAPK signaling pathway [105]. It is a common phenomenon that MAPK can be activated by upstream MPKK. 

Some studies have shown that MAPK can also be phosphorylated as the downstream of CDPKs. In rice, OsMPK5 could be activated by OsCPK4/18 [106]. In Arabidopsis, AtCPK5/6/11 could phosphorylate more MAPKs [71]. In maize, ZmCDPK11 was identified as the upstream of ZmMAPK5. When *ZmCDPK11* was transiently silenced in maize protoplasts, the expression and activity of *ZmMAPK5* decreased significantly. Both ZmMAPK5 and ZmCDPK11 could induce an increase in antioxidant protective enzyme SOD and APX activity. However, after transient silencing of *ZmMAPK5*, the increase in antioxidant protective enzyme activity was significantly weakened. This indicates that the regulation of antioxidant protective enzyme activity by ZmCDPK11 is likely to be achieved by regulating ZmMAPK5 [32]. Additionally, ZmCDPK11 could also regulate the expression of *ZmMAPK5*, probably implying that ZmCDPK11 could also regulate its transcription and activity. 

Moreover, there are other associations between ZmCDPKs and ZmMPKs. For example, ZmCPK15 interacted with ZmMPK3/4/6/7/8/9/11/17, ZmCPK17 interacted with ZmMPK3/6/13/14/19, ZmCPK28 interacted with ZmMPK2/4/8, and ZmCPK38 interacted with ZmMPK3/4 [27]. The interaction between these proteins in vitro has been verified, but the regulatory relationship is not clear. Interestingly, there was a complex regulatory relationship between ZmCPK38 homologue OsCPK18 and ZmMPK4 homologue OsMPK5. Either OsCPK18 could phosphorylate OsMPK5, or OsMPK5 could phosphorylate OsCPK18 in turn, indicating a mutual regulation between them [27,106]. These results suggest that there may also be bidirectional regulation and activation between CDPKs and MAPKs in maize, forming a complex signal network (Figure 1).

Besides MAPKs, there were other protein kinases that could also participate in the CDPK-regulated signaling pathway in plants under abiotic stress. Sucrose non-fermentative protein kinase SnRK is also a Ser/Thr protein kinase in plants. It is a major part of ABA signal transduction and is critical in plant signal transduction in response to stress [107]. There is communication between CDPKs and SnRKs in maize. The interaction between ZmCPK38 and ZmSnRK2.1, ZmCPK36 and ZmSnRK2.2, and ZmCPK17 and ZmSnRK2.5 has been proved in vitro [27]. Although studies have shown that there is a certain regulatory relationship between CDPKs, MAPKs, and SnRKs in maize, the relation of upstream or downstream signals and other specific regulatory mechanisms between CDPKs and these protein kinases still needs to be further explored.

## 9. Conclusions

ZmCDPKs in maize have been proved and analyzed to have many functions by many studies. ZmCDPKs mainly play a part in response to environmental stresses, such as drought, heat, and salt. CDPKs are often activated by calcium ions, transferring the phosphate group on ATP to the Ser/Thr site of the downstream protein, allowing the downstream protein to function. However, some CDPKs are not induced by Ca^2+^ and function independently of Ca^2+^, such as ZmCDPK1 [28]. CDPKs can respond to abiotic stresses in a variety of ways and exert kinase activity. The most important point is to act as a convergence point of different signaling pathways and transmit upstream signals to the downstream through phosphorylation. *ZmCDPKs* can be induced to express in various abiotic stresses. Subsequently, CDPKs can phosphorylate and activate proteins, such as transporters, HSPs, TFs, and other kinases. These proteins can improve/reduce the tolerance of plants to abiotic stress, so that CDPKs can complete signal transmission and play a positive or negative regulatory role. The summary and generalization of these proteins can provide better ideas for screening more CDPKs’ downstream proteins. 

In addition to the above pathways, the expression of *ZmCDPK11* in Arabidopsis inhibits the degradation of chlorophyll under salt stress and protects photosystem II from salt stress [31]; ZmCDPK12 contributes to the increase in chlorophyll content under salt stress [33]. Although these results indicate that CDPKs may also take part in the optical signal system, the specific regulatory mechanism is not detailed enough. The exploration of CDPKs’ functions in maize is not complete. Studies on other species have found that in addition to the S-type anion channel SLAC1, other ion channels are activated by CDPKs in response to abiotic stresses. The anion transporter SLAH3 can be activated by phosphorylation of AtCDPK2/3/6/20/21/23 [4]. Other transporters, such as the dual potassium channel TPK, can also be activated by phosphorylation of multiple CDPKs [108,109]. HSPs are an important component in plants’ response to abiotic stress, and the regulation of CDPKs on them is not accidental, but there are few studies on this. Many stress-related transcription factors also seem to have CDPK target sites and have been studied in other species, but there are few studies on the direct regulation of these transcription factors by ZmCDPKs in maize. There are also many regulatory relationships between other kinases and CDPKs, which deserve more attention. In addition, several studies have shown that CDPKs play a positive role in plant growth and stress response, but some CDPKs negatively regulate plant response to abiotic stress. For example, ZmCPK1 and ZmCPK25 negatively regulate maize response to cold stress [28]. However, the negative regulation of ZmCDPKs on abiotic stress still needs to be further analyzed.

There are various relationships, such as cooperation and antagonism, between proteins. There is complex crosstalk between signaling pathways, and the function of proteins in the pathway is not a single one. In higher plants, proteins can participate in different biological functions. With the in-depth study of CDPKs in maize, multiple signaling pathways have been discovered. The integration of these signal networks is a major challenge. Future research on maize stress tolerance will be aided by analyzing the function of ZmCDPKs involved in abiotic stress signal transduction.

## Figures and Tables

**Figure 1 ijms-24-02325-f001:**
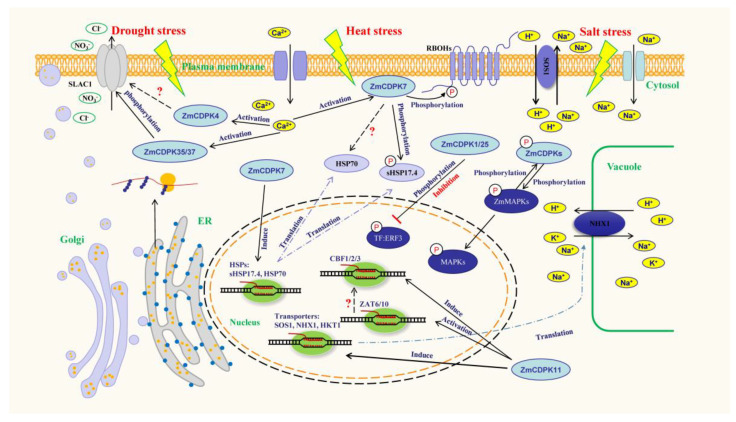
ZmCDPK signaling networks regulating maize abiotic stress responses. ZmCDPKs are important molecules involved in maize adaptation to drought, heat, salt, and other stresses. When these stresses occur, the intracellular Ca^2+^ concentration increases. ZmCDPKs are activated by Ca^2+^ and then phosphorylate downstream proteins to regulate gene expression or protein activity. ZmCDPKs phosphorylate ion channel protein SLAC1 to accelerate anion efflux and thus mediate stomatal closure. They can also regulate ion transporters SOS1 and NHX1 to regulate intracellular K^+^/Na^+^ ratio. The transcription of stress-related gene HSPs, transporters, CBFs, and ZATs can also be directly regulated or indirectly promoted by ZmCDPKs. The heat-shock protein sHSP17.4 can be activated by direct phosphorylation of ZmCDPK7, and the transcription factor activity of ERF3 was inhibited by phosphorylation of ZmCDPKs. Arrows indicate positive regulation; T-bars indicate negative regulation.

**Table 1 ijms-24-02325-t001:** 40 ZmCDPKs in maize.

Uniprot ID	Protein Names	Names in Article
A0A1D6MTQ3	Calcium-dependent protein kinase1	
A0A1D6EIK3	Calcium-dependent protein kinase2	
Q41790	Calcium-dependent protein kinase3	ZmCPK28 [19,27]
C0PHI0	Calcium-dependent protein kinase3	
C3UZ61	Calcium-dependent protein kinase4	ZmCPK3 [19]
C0P867	Calcium-dependent protein kinase5	ZmCDPK1 [19]
C0PEX3	Calcium-dependent protein kinase6	ZmCPK1 [28]
K7UI27	Calcium-dependent protein kinase7	ZmCPK5 [19]
B6SKK9	Calcium-dependent protein kinase7	ZmCPK37 [19,29] ZmCDPK7 [30]
B4FF99	Calcium-dependent protein kinase7	ZmCPK33 [19]
A0A1D6ICZ3	Calcium-dependent protein kinase8	ZmCPK25 [28] ZmCPK26 [27]
C0P4A7	Calcium-dependent protein kinase10	
Q7Y037	Calcium-dependent protein kinase11	ZmCDPK11 [29,31,32]
C0P875	Calcium-dependent protein kinase13	ZmCPK31 [19]
A0A1D6N998	Calcium-dependent protein kinase17	
A0A1D6F9X5	Calcium-dependent protein kinase20	
A0A1D6L6Z9	Calcium-dependent protein kinase20	
A0A1D6GFX7	Calcium-dependent protein kinase20	
A0A1D6ET07	Calcium-dependent protein kinase22	
A0A1D6I8I8	Calcium-dependent protein kinase24	
A0A1D6MUP6	Calcium-dependent protein kinase24	
A0A1D6IX89	Calcium-dependent protein kinase24	
B4FVF8	Calcium-dependent protein kinase28	ZmCPK38 [27]
K7U9U2	Calcium-dependent protein kinase28	ZmCPK39 [19]
A0A804RM42	Calcium-dependent protein kinase29	
A0A804M627	Calcium-dependent protein kinase30	
A0A1D6GKA4	Calcium-dependent protein kinase30	
C0HH11	Non-specific serine/threonine protein kinase	ZmCPK17 [19,27]
A0A804NZQ2	Protein kinase domain-containing protein	
C3UZ62	CDPK protein	ZmCPK12 [33]
K7V3X5	Putative calcium-dependent protein kinase family protein	ZmCPK14 [19,29]
B6U1G2	Calcium-dependent protein kinase, isoform AK1	ZmCPK35 [29]
K7VGN7	Putative calcium-dependent protein kinase family protein	ZmCPK29 [19]
A0A1D6G9R1	Putative calcium-dependent protein kinase family protein	
A0A1D6FSM1	Putative calcium-dependent protein kinase family protein	
K7VTH8	Putative calcium-dependent protein kinase family protein	ZmCPK22 [19]
B6UF43	Calcium-dependent protein kinase, isoform 2	
A0A804RGP8	Calcium-dependent protein kinase	
D87042	Calcium-dependent protein kinase	ZmCPK4 [34]

## Data Availability

Not applicable.

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
