# Peer review of "The Roles of CDPKs as a Convergence Point of Different Signaling Pathways in Maize Adaptation to Abiotic Stress"

_ijms, 2023, doi:10.3390/ijms24032325_

Round 1

Reviewer 1 Report

Title. The title is appropriate to the subject, informative, and concise.

Abstract. The abstract is concise, provides a clear overview, includes essential facts for the paper, and concludes with a final point that places the work described in a broader context.

Keywords. These are enough for the topic.

Review. The manuscript has an expend introduction, which includes background to provide an appreciation for the context of the work presented. Also, the review was correctly organized, the overall scope of the review was well-defined, and the integral parts were fitted together in logical order.

In the conclusion section, the authors included the major ones, which were briefly written.

Reviewer 2 Report

General

1.      As it is a review paper therefore, the title must reflect that it is a review paper. Therefore, please change the title to “The roles of CDPKs as a convergence point of different signaling pathways in maize adaptation to abiotic stress: A literature review”.

2.      Please write methodology for the study. How was literature reviewed? What was the criterion for literature selection and rejection? What types of key word were used? etc etc.

Abstract

Please re-write the abstract by giving background of the study, objectives/aims of the study, then outcomes and the conclusion you have concluded from the results.

Introduction

It is very brief and gap analysis is not given.

Activation and Expression of ZmCDPKs under Abiotic Stress

1.       Up to the present, 40 CDPKs have been found in maize. What are these? Please make a table of all these.

2.       Most ZmCDPKs function in a 70 calcium-dependent manner. Many studies have shown that the regulation of ZmCDPKs by Ca2+ is mainly determined by the protein structure of ZmCDPKs. What are these? Please make a table of all these.

ZmCDPKs Activate Transporters to Start Ion transport

1.       Under drought conditions, the process of CDPKs mediating ABA-regulated stomatal closure is likely to be achieved through the activation of ion channels, but whether ZmCDPK4 regulates ion channel proteins in mediating maize response to drought stress is not involved in current research. Please provide proper source citation/reference for it.

2.       Studies have shown that Na+/H+ antiporter in plasma membrane, the SALT OVERLY SENSITIVE 1 (AtSOS1), High Affinity Potassium Transporters (AtHKT1), Na+ , K+ , H+ antiporter and Na+/H+ exchanger 1 (AtNHX1) are important Na+/H+ antiporters in Arabidopsis, which play an important role in salt tolerance by regulating the content of Na+ and K+. How it occur? Please explain it through digram/figure.

ZmCDPKs Activate Transcription Factors to mediate Gene Expression

CDPKs in maize also seemed to activate some TFs. How it activate TFs? Please explain with the help of a diagram/figure.

Conclusion

1.      What are the benefits of this study?
